# Use of X-Irradiations in Reducing the Waste of Aflatoxin-Contaminated Pistachios and Evaluation of the Physicochemical Properties of the Irradiated Product

**DOI:** 10.3390/foods12163040

**Published:** 2023-08-13

**Authors:** Mohammad Hojjati, Samira Shahbazi, Hamed Askari, Mina Makari

**Affiliations:** 1Department of Food Science and Technology, Agricultural Sciences and Natural Resources University of Khuzestan, Ahvaz 63417-73637, Iran; 2Nuclear Agriculture School, Nuclear Science and Technology Research Institute (NSTRI), Atomic Energy Organization of Iran (AEOI), Karaj P.O. Box 31485-498, Iran

**Keywords:** nut, pistachio, mycotoxins, aflatoxin, malondialdehyde, antioxidant activity, protein

## Abstract

This study investigates the effects of electron beam irradiation (0, 1, 2, 4, and 6 kGy) on *Aspergillus flavus*, aflatoxin B1 (AFB1), and the physicochemical properties of pistachios. The findings suggested that e-beam significantly reduced the spore population of *A. flavus* and the concentration of AFB1 at doses of 4 and 6 kGy. Three AFB1 degradation products were detected via LC-MS analysis and their structures were presented. Total phenolic content was improved at a dose of 2 kGy, while antioxidant activity was decreased in all treatments in both DPPH and ABTS assays. The chlorophyll and carotenoid content declined and the color indices changed, leading to a darker color. E-beam at a dose of 2 kGy raised the soluble protein levels and changed the intensity and pattern of protein bands. Irradiation doses of up to 6 kGy enhanced the content of malondialdehyde and total saturated fatty acids while leading to a decline in unsaturated fatty acids. The quality features were adversely affected at doses > 4 kGy. The findings suggest that as an alternative method, e-beam at doses ≥ 2 kGy can effectively decrease fungal load and aflatoxin B1 contamination, and e-beam application at doses ≤ 2 kGy can maintain the physicochemical attributes of pistachios to an acceptable extent.

## 1. Introduction

*Aspergillus flavus*, belonging to *Aspergillus* genus section *Flavi*, is one of the most important molds producing toxic secondary metabolites known as aflatoxins [1,2]. Aflatoxins are the most common mycotoxins contaminating feed and food commodities during pre- or post-harvest stages and storage, leading to significant economic losses [3]. The most common aflatoxins produced by toxigenic strains of *A. flavus* are aflatoxin B1 (AFB1) and aflatoxin B2 (AFB2) [2]. AFB1 is more dangerous than AFB2 and other aflatoxins are. Due to its potent carcinogenic, hepatotoxic, teratogenic, and immunosuppressive impacts on humans, it is classified as a Group 1 carcinogen by the International Agency for Research on Cancer (IARC) [1,4]. The major genotoxic and carcinogenic metabolite of AFB1 is believed to be aflatoxin B1- exo-8,9-epoxide (AFBO), which is also metabolized into several oxidized metabolites such as hydroxylated products, including aflatoxin M1 (AFM1), aflatoxin Q1 (AFQ1), aflatoxin P1 (AFP1), and aflatoxicol (AFL). AFM1 is known to be the most carcinogenic hydroxylated metabolite of AFB1 [4]. Due to the high toxicity of AFB1 produced by *A. flavus*, numerous disinfection strategies containing physical, chemical, and biological methods have been applied to decrease or eradicate it from different types of food products and reduce its potential health risks. Although these treatments can be used to diminish fungal contamination and the deleterious effects of AFB1, they require complicated equipment and are not economically efficient [5]. Additionally, the chemical detoxification of products intended for human consumption is not permitted in the European Union (EU) [3]. Over the past decades, electromagnetic radiation has been studied as an alternative disinfection method to eliminate *A. flavus* and reduce or remove AFB1 contamination from foodstuffs [2,3,5,6,7,8,9,10,11].

Ionizing radiation has been known as an efficient decontamination technique as it extends the shelf life of fresh food products without negatively affecting their nutritional value and quality features [12]. According to the FAO/IAEA/WHO Joint Expert Committee, food irradiation up to 10 kGy does not pose any microbial, nutritional, or toxicological hazards [13]. Food irradiation has been extensively used in Asia and the United States [14]; however, the EU has reduced commercial food irradiation after the EU directive 1999/2/EC [15]. Moreover, EU regulation 1999/3/EC permitted the application of ionizing irradiation to a few food commodities comprising spices, dehydrated aromatic herbs and vegetable seasonings [16]. Nonetheless, EU members are still allowed to utilize irradiation for different types of foodstuffs, especially nuts [17].

Food irradiation consists of subjecting food to a non-thermal ionization source, including gamma rays (Co^60^ or infrequently by C^137^), electron beams (e-beams) with energies less than 10 MeV, and X-rays with energies less than 5 MeV [18]. Γ-irradiation using a cobalt-60 radioisotope is the preferred food irradiation technique due to its high penetration, allowing the treatment of foodstuffs with less handling. Nevertheless, due to the continuing decay of radioactive energy, the Co 60 source needs to be regularly recalibrated and cannot be switched on and off [19]. X-rays have a low penetration ability and are generated from electrical machine sources of energy that utilize target materials such as gold or tantalum [18]. High-energy electron beams are generated by electron-accelerating machines as the source of energy, converting energetic electrons into electromagnetic X-rays known as bremsstrahlung [12,18]. E-beams are ionizing radiation with limited penetration mainly used in foods at energy levels up to 10 MeV, reducing the degradation of food products [12]. The application of e-beams as a food pasteurization technique has several advantages over γ-irradiation, including the fungal decontamination of food, improved shelf life of food products, easy operation, lack of charging requirements for the source, abundant accessibility, lack of radioactivity in food, cost-effectiveness, and safety [2,12,18].

Irradiation affects molds via direct and indirect mechanisms of action. The high ionization energy directly damages the genetic material of the microorganism and causes mutation, leading to cell lesions or even death [7]. Water radiolysis during irradiation creates highly reactive free radicals, which indirectly damage microorganisms and result in aflatoxin degradation [9]. Several studies have suggested that ionizing radiation can effectively reduce fungal and aflatoxin (AFB1 and AFB2) contamination in various food products. Γ-irradiation (10 kGy) reduced the AFB1 concentration by about 58%, 69%, and 88% in peanuts, pistachios, and corn, respectively [3]. A 45% reduction in AFB1 concentration was achieved after gamma irradiation (10 kGy, 10 min) in hazelnuts [5]. Γ-treatment of maize at doses of 2, 5, and 10 kGy decreased the number of *A. flavus* spores under the detection limit and degraded AFB1 levels by 46% and 68.9%, and AFB2 levels by 97.6% and 94% at doses of 2 and 5 kGy, respectively [11]. The number of *A. flavus* spores was reduced under the investigation limit, and AFB1 levels were diminished to 73.26% and 83.36% at γ-treatment doses of 4 and 6 kGy in pistachios [6]. Jalili et al. (2012) [8] reported that γ-treatment at a dose of 6 kGy of contaminated red chilies showed an 86% to 98% decrease in the AFB1 concentration and a reduction in the colony count of *A. falvus*.

Pistachio (*Pistacia vera* L.) is an agricultural produce highly prone to toxigenic *A. flavus* [20]. Based on European Commission regulatory requirements, the allowed levels of AFB1 and total aflatoxin in pistachios are restricted to 8 μg/kg and 10 μg/kg, respectively [21]. Pistachio is an abundant source of bioactive constituents including unsaturated fatty acids, sterols, polyphenols, antioxidants, vitamins, proteins, and minerals [20,22]. As with any food processing technique, irradiation can cause unwanted physical or chemical changes. Although the irradiation of food products up to 10 kGy induces no nutritional problems [18], pistachio is abundant in unsaturated fatty acids and proteins that are targets of hydroxyl and superoxide free radicals generated during irradiation, leading to oxidation and peroxidation [20,23]. High doses of radiation can also induce undesirable changes in the sensory attributes (flavor, texture, and appearance) of pistachio nuts, making them unacceptable for human consumption [24]. Therefore, it is necessary to assess the effects of e-beam irradiation on the physicochemical and quality attributes of irradiated pistachios.

This study aimed to evaluate the effect of electron beams’ dose range on the inactivation of *A. flavus*, reduction in AFB1 levels and its degradation products, and some physicochemical and sensory attributes of pistachios.

## 2. Materials and Methods

### 2.1. Sample Preparation

Pistachio nuts (*Pistacia vera* L.), Akbari cultivar, were collected from the Iranian Pistachio Research Center located in Rafsanjan, Kerman province, Iran, then transferred to the laboratory of the Nuclear Agriculture Research Institute in Karaj, Alborz province, Iran. All raw in-shell pistachio nuts were screened for skin uniformity, healthiness, and open shells, then put in polyethylene bags and stored in a refrigerator (4 ± 1 °C) until electron beam irradiation and further tests were performed.

### 2.2. Fungal Inoculation and AFB1 Contamination of Pistachio Nuts

The aflatoxigenic strain of *Aspergillus flavus* obtained from the Iranian Research Organization for Science and Technology (IROST) was cultured on potato dextrose agar (PDA, Merck, Darmstadt, Germany) and kept at 28 °C for five days. Some pistachio kernels were placed on the culture medium containing pure *A. flavus* spores, then incubated for two days at 28 °C so that the spores were well-transferred to the surface of the pistachio kernels. The contaminated pistachios were ground with the rest of the pistachios, and a suspension of 0.5 g of infected powdered pistachio nuts was prepared in 5 mL of physiological serum comprising 0.5% Tween-80. The number of spores was counted utilizing a hemocytometer, and the population of spores in infected pistachios was set to (6.18 ± 1.18) × 10^5^ spores/g of pistachio nuts. Finally, 100 g of contaminated pistachios was placed in polyethylene bags and kept at 4 ± 1 °C until e-beam irradiation was performed [5].

In-shell pistachio nuts were contaminated with pure AFB1 obtained from *A. flavus* (Sigma-Aldrich, Darmstadt, Germany) diluted in acetonitrile aqueous solution by spraying on their surface so that the final concentration of the toxin in the samples was 600 ppb. Then, in each repetition of each treatment, 100 g of contaminated pistachio samples was placed in polyethylene bags for e-beam irradiation [6].

### 2.3. Electron Beam Irradiation

Pistachio samples contaminated with *A. flavus* and AFB1 were exposed to 0, 1, 2, 4, and 6 kGy dosages of e-beams provided by the electron accelerator (10 MeV) located in Taft, Yazd province, Iran, affiliated with the Organization Atomic Energy with a maximum beam current of 3 mA and an effective beam width of 25 mm. The samples were passed in front of the electron accelerator on the automatic conveyor controlled by a computer. A Red 4034 Perspex dosimeter from Harwell Technologies (UK) was employed to monitor the absorbed dose. All treatments were performed in 3 replicates.

### 2.4. Fungal Load Determination

Spores were counted by preparing a suspension of e-beam-treated pistachios in sterile saline (1000 mL sterile distilled water containing 8.5 g sodium chloride) (in 10^−6^ dilutions utilizing a hemocytometer). The propagule and spore population of *A. flavus* were determined by preparing serial dilutions in sterile saline and spread-plating each dilution (100 µL) onto the surface of the PDA agar (Merck, Darmstadt, Germany) medium supplemented with 50 ppm of chloramphenicol. Then, colonies were counted after incubating inoculated Petri dishes at 28 °C for 3 to 5 days. The counts of colonies obtained from the triplicates were reported as logarithmic colony-forming units per gram (log cfu/g) [6].

### 2.5. Aflatoxin B1 Extraction and Detection by HPLC

The levels of AFB1 were evaluated using an HPLC device, model 1220 of Infinity Isocratic LC, manufactured by Agilent based on the method of Makari et al. (2021). For HPLC analysis, a mixture of pistachio nut samples (1 g) with methanol (20 mL of 80%) was prepared and shaken thoroughly (at 150 rpm for 24 h). The centrifugation of the extract was performed (at 1000× *g* for 5 min) and then the supernatant was gathered and passed through an ASPEC 401 immunoaffinity column containing AFB1 monoclonal antibody at a speed of 2 mL/min. The immunoaffinity column was washed with phosphate-buffered solution (PBS) (10 mL), filled with the supernatant, and washed again with 10 mL of distilled water to remove impurities. The bounded AFB1 was released by eluting it with acetonitrile (1.5 mL) for 60 s. Eventually, before HPLC analysis, 0.5 mL of the collected eluate was diluted with 2 mL of water, and 400 μL of it was injected into an HPLC system coupled with a reversed-phase C18 column (Spherisorb Excel ODS1) (dimensions 4.6 × 250 mm and particle diameter five μm), a column guard (25 × 4.6 mm internal diameter), and a fluorescence detector (Perkin Elmer LC420) with emission at 440 nm and excitation at 364 nm. Water: acetonitrile: methanol acetonitrile (56:30:14 *v*/*v*/*v*) at a flow rate of 0.86 mL/min was utilized as the mobile phase. A post-column in a zero-dead-volume T-piece and polytetrafluoroethylene (PTFE) reaction tube (0.3 mm × 30 cm id.) at 70 °C was used for derivatization. The derivatization reagent pyridine hydrobromide perbromide (PBPB) was used at a 0.3 mL/min flow rate. The retention time of AFB1 was around 13.37 min. The HPLC system was validated by drawing a calibration curve utilizing a series of standard solutions in triplicates [9].

### 2.6. LC-MS Analysis of Degraded AFB1

For liquid chromatography–mass spectrometry (LC-MS) analysis, AFB1 was extracted from the samples using an 80% ethanol aqueous solution. AFB1 degradation products were detected by injecting 10 µL of the extract into an Agilent 1200 LC-MS system (Agilent Technologies, Inc., Santa Clara, CA, USA) equipped with an Agilent Eclipse Plus C18 column (100 mm × 3.0 mm with a 1.8 μm particle size) and a 6410 triple-quadrupole mass spectrometer with a flow rate of 0.3 mL/min at 60 °C. The mobile phase contained 0.1% aqueous formic acid (component A) and acetonitrile with 0.1% formic acid (component B). The analysis was performed in both gradient and isocratic elution modes. The procedure began the gradient for 1 min with 10% of B, and then B was escalated to 95% for 20 min and kept isocratic for 7 min. The analysis was performed in triplicates within a total run time of 30 min per sample. The MS was performed in positive-ion mode with the following conditions: a capillary voltage of 3.2 kV; desolvation temperature of 350 °C; nebulizing gas flow of 3 L/min; source block of 110 °C; drying gas flow of 20 L/min; and mass range of 200–400 *m/z* [25].

### 2.7. Determination of Total Phenolic Content (TPC) and Antioxidant Activity (AOA)

Total phenolic content (TPC) was measured via the Folin–Ciocalteu assay [22], and antioxidant activity (AOA) was quantified using DPPH [6] and ABTS [26] free radical scavenging activity assays. The extraction was done by mixing 1 g of powdered pistachios with 30 mL of 80% methanol solution. Mixture sonication (Trans Sonic TP 690-A, Elma, Singen, Germany) was performed twice for 15 min, and then the mixture was kept at room temperature for a day. The mixture was centrifuged (Hettich Refrigerated Centrifuge Universal 320R, Tuttlingen, Germany) for 10 min at 1500× *g* and 4 °C. The collected supernatant was used to measure the TPC and AOA.

### 2.8. Determination of Chlorophyll and Carotenoid Pigments

The measurement of chlorophyll and carotenoid pigments of pistachio nuts was carried out using the spectrophotometric method [22]. An amount of 2 g of pistachios was mixed with 5 mL of 80% acetone solution (*v*/*v*) and kept for 15 min in the dark at ambient temperature. After the centrifugation (at 1500× *g* for 5 min) and filtration of the extract, the absorbance was quantified at 663 (A663), 645 (A645), and 470 (A470) nm. The chlorophyll a (C_a_), chlorophyll b (C_b_), and carotenoid (C_t_) contents were determined ((1)–(3)).
C_a_ = 12.21 (A_663_) — 2.81 (A_645_)(1)
C_b_ = 21.13 (A_645_) — 5.03 (A_663_)(2)
C_t_ = [1000 (A_470_) — 3.27C_a_ — 104C_b_]/229(3)

### 2.9. Color Analysis

The color indices of pistachios, including *L (brightness–darkness), a* (redness–greenness), and b* (yellowness–blueness), were determined using a Minolta colorimeter CR-400 (Konica Minolta. Inc., Osaka, Japan). To make the test uniform and reduce data changes, about 10 g of powdered pistachio from each sample was placed on a small Petri dish, and a watch glass was placed on it. The color indices were recorded from five different angles of each sample using the colorimeter probe. The total color difference (ΔE) values of the treated samples were calculated using the following Formula (4) [27]:
(4)$$\Delta E=\sqrt{{\Delta L}^{2}+\Delta a^{2}+\Delta b^{2}}$$


### 2.10. Total Soluble Protein (TSP)

Total soluble protein (TSP) was assessed according to the Bradford assay [28]. First, a mixture of pistachio samples and 10% sodium phosphate-buffered solution (pH = 6.8, *w*/*v*) comprising glycerol was stirred for 30 min and centrifuged (5000× *g* for 5 min). Then, the supernatant was collected and kept at −20 °C for further experiments. Next, about 100 μL of the obtained extract was transferred to a test tube containing 5 mL of Bradford reagent and held at ambient temperature for 30 min. Finally, the absorption was quantified with the help of a BSA calibration curve at a wavelength of 595 nm. Total soluble protein results were expressed in mg of protein per g of pistachios (mg/g).

### 2.11. Native-PAGE and SDS-PAGE Experiments

Native-PAGE and SDS-PAGE were conducted using 5% stacking gel and 14.5% separating gel in Bio-Rad Mini-Protean II Dual Slab Cell (Hercules, CA, USA) based on the Laemmli method [29]. To prepare the protein sample, 1 g of pistachio kernels was powdered, washed, defatted twice with acetone, and homogenized (at 7000 rpm for 5 min) using 10 mL of 100 mM Tris buffer, at pH 6.8. Then, protein concentration was measured using Bradford’s reagent [28]. About 300 µg from each acetone-precipitated sample was collected via centrifugation (at 4500 rpm for 7 min). After removing the acetone from the precipitated samples, 100 µL of double distilled water was added and stirred thoroughly. An amount of 100 µL of the sample buffer was added and placed in a boiling water bath for 5 min. Next, 20 µL of the samples was loaded onto the gel wells. Finally, the gel was stained using Coomassie Brilliant Blue R-250 and destained in methanol: acetic acid: water (8:1:1 *v*/*v*). Gel images were recorded using GelDoc XR+ (Bio-Rad Laboratories, Inc., Hercules, CA, USA), and band density was assessed utilizing the Quantity One 1-D Analysis software. Molecular weight, relative density, and band intensity were quantified using Gel-Pro Analyzer (ver.6.0, Media Cybernetics, Silver Spring, MD, USA).

### 2.12. Determination of Malondialdehyde (MDA)

Malondialdehyde (MDA) was measured using the thiobarbituric acid reactive substances (TBARS) assay [6]. Approximately 7 g of the pistachios was mixed with 15 mL of 7.5% trichloroacetic acid solution (TCA) (*w*/*v*), including 0.1% EDTA (*w*/*v*) and 0.1% propyl gallate (*w*/*v*), then homogenized (18,000 rpm for 1 min) and adjusted to a volume of 30 mL using TCA solution (150 mm filter paper). The extract was filtrated, and about 2.5 mL of it was transferred into a test tube containing 2.5 mL of 46 mM TBA reagent in 99% TCA, then heated in a water bath for 35 min and chilled in a cold-water bath. The absorbance of the samples was read using a spectrophotometer at a wavelength of 532 nm.

### 2.13. Analysis of Fatty Acid Composition

The fatty acids of the un-irradiated (non-irradiated) and irradiated pistachio nuts were extracted and evaluated based on the method of Mexi and Kontominas (2009) [20] with slight changes. Concisely, about 0.05 g of extracted oil from pistachio nuts using a Soxhlet extractor with di-ethyl ether was saponified using 100 μL of dichloromethane and 1 mL of methanolic sodium hydroxide solution (10 min at 90 °C). Then, samples were esterified by adding 1 mL of a 14% methanolic mixture of bromine trifluoride (BF3-methanol) to the above solution and placed in a boiling water bath (10 min). About 600 μL of n-hexane was added to separate the fatty acid methyl esters from the above mixture. To identify and quantify fatty acid methyl esters, 1 μL of the samples was fed to an Agilent model 6890A gas chromatography machine coupled to a mass spectrometry detector (MSD 5975, Agilent Technologies). The injection chamber and the detector were stabilized at 230 and 290 °C, respectively. Fatty acid separation was performed using helium as a carrier gas with a flow rate of 1 mL/min in a split ratio of 1:20 on a capillary column, HP-5MS (30 m × 0.25 mm id., 0.25 μm film thickness). The column temperature was initially set at 60 °C and held for 2 min, then increased to 170 °C at a rate of 3 °C /min and kept for 10 min. Then, it was maintained and increased to 220 °C at a rate of 10 °C/min and held for 10 min. Electron impact ionization was performed at 70 eV, and mass spectra were acquired in a mass range of 35–450 *m/z*. Fatty acid identification and measurement were carried out utilizing the area under the curve and using fatty acid standards.

### 2.14. Sensory Evaluation

Sensory attributes of un-irradiated and irradiated pistachio nuts, including color, crispiness, taste and odor, and overall acceptability, in the framework of the nine-point hedonic scale, were evaluated by seven trained panelists including four women and three men in the age range of 25 to 47 years old who had sufficient experience in evaluating food products, including nuts. The samples were prepared in similar conditions and given to the panelists for scoring. For sensory evaluation, five pistachios from un-irradiated and irradiated samples were placed in a small plastic container with a random three-digit code in a suitable environment for panelists to score according to the paper ballot in front of them and rate the desired features from 1 to 9. The scores of 1 and 9 were recorded as the lowest and highest for each sample, respectively [30].

### 2.15. Statistics and Data Analysis

A factorial analysis of variance (ANOVA) was performed in the form of a completely randomized design. Duncan’s multiple-range test at a confidence level of 95% was performed for a comparison of means. Data analysis was performed using SPSS Software v 23.0 (SPSS Science, Chicago, IL, USA). All the tests were performed at least in triplicate.

## 3. Results and Discussion

### 3.1. Inactivation of A. flavus by Electron Beam Irradiation

Figure 1 shows the results of different doses of e-beam (0, 1, 2, 4, and 6 kGy) on the *A. flavus* spore population in contaminated pistachio nuts. With an increase in the e-beam dose, a significant decrease was observed in the population of *A. flavu*s spores (*p* < 0.5). The initial contamination of pistachio nuts was 4.620 log cycles. The spore population decreased by 1.491 and 3.868 log cycles at doses of 1 and 2 kGy, respectively, while increasing the irradiation dose up to 6 kGy reduced the population of spores by 4.620 log cycles. Therefore, the e-beam doses of 4 and 6 kGy were the effective doses with which to successfully remove *A. flavus* spores from infected pistachio nuts.

Molds are prone to ionizing radiation and can be damaged directly or indirectly via high-energy photon exposure. Fungal spores can be damaged directly through ionization, leading to lesions in the genetic material of the cells, a breakage of both DNA strands, mutations, and even cell death depending on various factors [7,9]. The indirect damage of irradiation is due to the radiolysis of water, which induces the formation of different reactive particles, including active oxygen species (ROS), free radicals, and peroxides [9]. The lethal dose of the e-beams on microorganisms relies on several factors, including irradiation (the rate of absorption and the total absorbed dose), ambient conditions (temperature, humidity, and air composition) throughout the process, and microorganism sensitivity to irradiation [12]. The radioresistance of fungi is more than that of vegetative forms of bacteria and yeasts due to the presence of melanin in their hyphae and low DNA content [6,12]. Thus, the irradiation dose for controlling the contamination and disinfection of food products depends on the type and microbial strain, stage of microbial growth, and number of microorganisms [6].

These results are also in accordance with our earlier observations, which showed that γ-irradiation at doses ≥ 4 kGy effectively diminished the population of *A. flavus* spores in infected pistachios [6]. A recent study reported the complete prevention of the spore germination of A. flavus after γ-irradiation of contaminated maize samples at a dose of 6 kGy [7]. *A. flavus* was eliminated after the application of γ-rays and e-beams at doses of 5 and 10 kGy in Brazil nut [2]. According to Markov et al. (2015) [1], γ-irradiation (5 kGy) suppressed the germination, sporulation, and growth (up to 90%) of aflatoxigenic *A. flavus*, *A. parasiticus*, and *A. niger* in pure or mixed cultures. Aquino et al. (2005) [11] demonstrated that the spore population of *A. flavus* decreased in maize after γ-irradiation. Iqbal et al. (2013) [9] observed a decrease in the population of *A. flavus* spores at higher doses of γ-irradiation (6 kGy) in ground and whole red chilies. Boonchoo et al. (2005) [10] detected a decline of 80% in the population of *A. flavus* spores in contaminated brown rice after γ-irradiation (6 kGy). According to Byun et al. (2019) [27], the spores of *A. flavus* decreased in red pepper powder and red chili paste with increasing irradiation doses of γ-ray, X-ray, and e-beam up to 3.5 kGy.

### 3.2. Aflatoxin B1 Degradation

Table 1 illustrates AFB1 concentrations in un-irradiated and irradiated pistachio samples at different e-beam doses of 0, 1, 2, 4, and 6 kGy. By increasing the dose of the e-beam, the concentration of AFB1 significantly decreased in contaminated pistachio nuts (*p* < 0.05). A direct correlation was observed between the level of AFB1 breakdown and the increase in the irradiation dose [3,6,9,11]. The AFB1 concentration declined by 21.1% and 75.65% at doses of 1 and 2 kGy, respectively (Table 1). E-beam irradiation doses of 4 and 6 kGy showed the highest decrease in the levels of AFB1, which were 99.69% and 100%, respectively. The chromatograms presented in Figure 2 display the effect of different doses of e-beam on pistachio samples contaminated with AFB1. According to the results of the chromatograms, the peak of AFB1 was detected in the retention time of 13.37 min in the control sample and the irradiated samples at doses of 1, 2, and 4 kGy. Nevertheless, no peak related to AFB1 was observed in the chromatogram obtained from the irradiated sample at a dose of 6 kGy at a retention time of 13.37 min.

The same results were obtained from previous studies reporting that AFB1 concentration was significantly decreased after γ-irradiation (4 and 6 kGy) by approximately 73.27% and 86.36%. [6]. Sen et al. (2019) [5] observed that the AFB1 concentration in hazelnuts declined by about 47% after γ-irradiation at a dose of 10 kGy for 10 min. According to Ghanem et al. (2005) [3], γ-rays (of 4, 6, and 10 kGy) remarkably reduced the AFB1 concentration in peanut, pistachio, rice, corn, and feed (wheat, wheat bran, and corn). In the research by Assuncao et al. (2015) [2], reductions of 53.32% (5 kGy) and 65.66% (10 kGy) in AFB1 concentration were reported after the e-beam radiation of Brazil nut. Additionally, γ-rays lowered the AFB1 concentration in Brazil nut by 70.61% and 84.15% at the same doses [2]. A reduction of about 68.9% and 47% in AFB1 concentration was reported after γ-treatment at doses of 2 and 5 kGy in infected maize samples [11]. Jalili et al. (2012) [8] reported that γ-irradiation (30 kGy, 13% moisture) of AFB1-contaminated black and white peppers reduced the toxin concentration by roughly 50.6%. Previous studies have noted an inverse correlation between the foods’ oil content and levels of AFB1 degradation. Aflatoxin degradation was more complex in food containing oil (pistachio nuts, peanuts, hazelnut, and corn) than in pure aflatoxin aqueous solutions [3,5]. The amount of water in food plays a crucial role in AFB1 degradation. Water radiolysis generates highly reactive free radicals that can attack double bonds, especially those in aromatic or heterocyclic rings or the carbonyl group of lactone and furan rings, destroying the molecular structure and lowering the toxigenicity of AFB1 [8].

### 3.3. Aflatoxin B1 Degradation Products

AFB1 and its degradation products formed under e-beam irradiation (0, 1, 2, 4, and 6 kGy) were identified with the help of LC-MS. Chromatograms of AFB1-contaminated pistachios treated with e-beam (0, 1, 2, 4, and 6 kGy) are shown in Figure 3. Three products of AFB1 degradation, including C1, C2, and C3, were detected in the e-beam-treated samples, which were absent in the control sample (Figure 3). The absence of the degraded products in the control sample suggested that these products were generated from AFB1 degradation. The comparison of the peaks identified from the LC analysis revealed that the peak areas of AFB1 decreased, and the peak areas of degradation products increased with the rise in the radiation dose. Additionally, the peak area related to C1 decreased at doses ≥ 2 kGy, and the peak areas related to C2 and C3 increased at the same doses, suggesting that the degradation product C1 was an intermediate reaction product that converted into degradation product C3 at higher doses. AFB1 degradation products developed at e-beam different dosages were detected utilizing MS analysis, and their proposed structures are shown in Figure 4. As can be seen in Figure 4, the AFB1 (312.105 *m/z*) structure transformed into its three degradation products, including C1 (328.06 *m/z*), C2 (318.11 *m/z*), and C3 (347.20 *m/z*), under e-beam treatment. E-beam treatment cleaved the furan ring (C2), modified lactone and benzene rings, and hydroxylated cyclopentenone (C3 and C1, respectively).

The possible degradation pathway for AFB1 and the molecular formulas of its degradation products are illustrated in Figure 5. According to the proposed pathway, the degradation of AFB1 due to e-beam irradiation started with the hydroxylation of cyclopentenone and the formation of C1 as an intermediate compound. It should be noted that C1 showed a structure similar to that of aflatoxin M4 (AFM4), the hydroxylated metabolite of AFB1 with less toxicity [31]. Further, increasing the radiation dose led to lactone ring cleavage, adding a hydroxymethyl and a hydroxyl group to the cyclopentanone, the hydroxylation of the benzene ring, and the formation of the degradation product C3. On the other hand, the cleavage of the furan ring and the breakage of the C8–C9 double bond, and the addition of two aldehyde groups, following the deprotonation of the benzene ring and phenolate formation, led to the development of C2.

AFB1 toxicity and carcinogenicity are closely related to its structure. The C8–C9 double bond on the furan ring and the lactone ring are the main toxicological active sites of AFB1 [32]. Accordingly, any alteration in these active sites would contribute to a reduction in the bioactivity of AFB1. Figure 5 shows the structures of two main AFB1 degradation products (C2 and C3) that were changed due to the cleavage of the furan C8–C9 double bond (C2) or the cleavage of the lactone ring and hydroxylation of the cyclopentenone group (C1 and C3). Thus, the degradation products appear to be less toxic than AFB1. According to Liu et al. (2018) [33], AFB1 degradation products in e-beam-irradiated peanut meal showed lower mutagenicity and cytotoxicity than the un-irradiated ones did. E-beam treatment diminished the toxicity and mutagenicity of AFB1 in an aqueous medium [32]. Biological toxicity tests for AFB1 degradation products were not performed in this study. However, conducting these tests to confirm the safety of degradation products is necessary and will be considered in future studies.

### 3.4. Total Phenolic Content (TPC) and Antioxidant Activity (AOA)

Table 2 summarizes the effects of e-beam irradiation on the TPC and the AOA of the treated pistachio nut kernels via two methods of inhibition of DPPH and ABTS free radicals. All the data differed significantly (*p* < 0.05). As presented in Table 2, e-beam progressively improved the amount of TPC at doses ≥ 2 kGy. The highest and lowest amounts of TPC were detected at doses ≤ 2 kGy, respectively. The results showed a decline in the AOA of treated pistachio nut kernels with the increasing dose of e-beam radiation with both DPPH and ABTS methods. The highest AOA decrease was detected at the dose of 6 kGy with both methods, while the lowest decline in AOA was observed with the DPPH method at a dose of 1 kGy. The AOA of pistachio samples at a dose of 1 kGy remained unchanged in the ABTS method and did not show any significant difference at doses of 2 and 4 kGy (*p* > 0.05).

The observed increase in the TPC of irradiated pistachio nuts can be attributed to the changes in enzymes’ catalytic activity associated with the biosynthetic pathway of the phenylpropanoid or phenylalanine ammonia-lyase activity improvement caused by low irradiation doses [34,35]. These results are in accord with those of recent studies indicating that doses ≤ 2 kGy enhanced the TPC of irradiated pistachio nuts [6,22,34]. Dixit et al. (2010) [36] also reported that γ-rays (2 kGy) increased the TPC of soybeans. According to Kim et al. (2010) [35] γ-rays (0.5 to 2 kGy) increased the TPC of irradiated peaches. Like our results, a reduction in the AOA of γ-irradiated pistachio nuts was reported at high doses of γ-irradiation (≥4 kGy) [6,22,34]. The reduction in AOA attributes at higher doses of ionizing radiation may be related to the detrimental effects caused by generating a flux of free radicals [36]. In contrast with our findings, the literature review showed an improvement in the AOA of pistachio nuts and soybean with low irradiation doses (≤2 kGy) [6,22,34,36]. Some authors have also suggested that AOA enhancement via irradiation may contribute to free phenolic compounds being released from the food matrix containing flavonoids as a result of the breakdown of glycosides when exposed to ionizing radiation, the upregulation of pathways engaged in antioxidant defenses, a TPC level increase, or genetic variation [34,36].

### 3.5. Chlorophyll and Carotenoids Content

The results of e-beams (0, 1, 2, 4, and 6 kGy) on the pigment content of pistachios are listed in Table 2. There was a reduction in chlorophyll a and b content when increasing the dose of the e-beam in all pistachio nut samples. The highest decline in chlorophyll a and b content was detected at absorbed doses ≥ 4 kGy, and was about 48.2% and 42.5%, respectively. E-beam absorbed doses of 1 and 2 kGy led to the lowest decline in chlorophyll a content (25.8% and 23.7%, respectively). However, the amount of chlorophyll b remained unaffected at the same absorbed doses. In the same way, total chlorophyll was significantly diminished by increasing the e-beam dose (*p* < 0.05). The amount of total chlorophyll in the e-beam-treated pistachios at doses of 4 and 6 kGy was reduced by 40.74% and 41.04%, respectively. Total chlorophyll was in higher amounts at doses of 1 and 2 kGy compared to that of the control sample. The chlorophyll content reduction in treated samples may have been due to the high doses of e-beam irradiation damaging the chlorophyll pigments of the pistachio samples. This effect can also be attributed to the alterations in the cellular structure and metabolism such as the dilation of the thylakoid membranes, a change in photosynthesis, the modulation of the antioxidative system, and the cumulation of phenolic compounds. A possible explanation for chlorophyll b reduction via irradiation might be the deterioration of chlorophyll b precursors or selective destruction of its biosynthesis [37]. As indicated in Table 2, the total carotenoid content declined with an increasing the radiation dose. The highest reduction in carotenoids (32.54% and 27.78%) was detected in treated pistachios at absorbed doses of 1 and 6 kGy, and the lowest reduction (21.77% and 25.56%) was at doses of 2 and 4 kGy, respectively. The carotenoid reduction caused by e-beam treatment might be ascribed to increased oxidative reactions and the formation of secondary free radicals (^•^OH, H_2_O_2_, and O_3_) through irradiation [38]. A loss of carotenoids at irradiation doses > 1 kGy can also occur due to the alteration of the biosynthetic pathway of carotenoids [39]. Overall, chlorophyll and carotenoid reduction due to e-beam irradiation can lead to the discoloration of pistachios.

These results are consistent with the findings of Alinezhad et al. (2021) [22] and Makari et al. (2021) [6] which showed that γ-irradiation at doses ≥ 4 kGy decreased the content of chlorophyll a, chlorophyll b, total chlorophyll, and carotenoids in pistachio nuts. Ramamurthy et al. (2004) [40] observed a decline in the chlorophyll content of capsicum via γ-rays (1, 2, and 3 kGy).

### 3.6. Color

The results of different doses of e-beam irradiation on L*, a*, and b* color indices and the total color difference (ΔE) of pistachios are displayed in Table 2. As illustrated in Table 2, low doses of e-beam (≤2 kGy) did not change the L* parameter or the brightness of the samples, whereas higher doses (≥4 kGy) reduced the values of L*, causing the samples to develop a darker color. Conversely, irradiation at a dose of 6 kGy raised the a* parameter or redness of the samples. The values of the a* parameter at other doses did not significantly differ from those of the control sample (*p* > 0.05). The b* parameter was reduced at doses ≥ 4 kGy and tended towards the blue spectrum, but the b* parameter did not show any changes when exposed to other e-beam doses. Amongst the irradiated samples, the total color difference (ΔE) evaluation showed that the color change was 0 < ΔE < 0.5 and not distinct in the irradiated samples at doses of 1 and 2 kGy. However, the color change of ΔE > 1.5 was distinct in the irradiated pistachios at doses ≥ 4 kGy [27] General variations in color parameters led to the darkening of irradiated pistachios. This color alteration of darkening in e-beam-irradiated pistachio nuts can be explained by the breakdown of products such as carbonyl and amino compounds attributed to the disruption of glycosidic and peptidic bonds, resulting in the Maillard reaction and formation of colored compounds. Another reason for the darkening in irradiated pistachio nuts may be the oxidation of phenols [20].

These results are in line with those of previous reports suggesting that the indicated γ-irradiation (0.5, 1, 1.5, 2, 4, and 6 kGy) altered the L*, a*, and b* color indices and darkened the irradiated pistachio nuts [6,22]. In two separate studies, Mexis and Kontominas (2009) [20,30] reported an increase in the a* color value of pistachio and cashew nuts at higher doses of γ-irradiation, which darkened both nuts. Moreover, the L* and b* color parameters of γ-irradiated (0.5, 1, 3, and 5 kG) pine nuts were reported to be remarkably different (*p* < 0.05) [41]. Sánchez-bel et al. (2008) [42] observed a slight darkening of almonds after e-beam irradiation (3, 7, and 10 kGy). In contrast, Güler et al. (2017) [43] observed that the color of hazelnut kernels remained unaffected after γ-irradiation (0.5, 1, and 1.5 kGy).

### 3.7. Solubility of Proteins (TSP)

Solubility is the most commonly applied criterion of protein aggregation, which is affected by primary amino acid composition, molecular weight, and polypeptide structure, and is a suitable indicator of protein functionality [44]. Table 2 depicts the effects of e-beam treatment on the soluble protein content of pistachio samples. The results demonstrated a remarkable increase in the TSP of pistachio nut kernels at e-beam doses of 1, 2, and 6 kGy. Irradiation at a dose of 2 kGy displayed the highest TSP (*p* < 0.05). Nonetheless, the TSP of the pistachio kernels remained unaltered at the absorbed dose of 4 kGy. Ionizing radiation has been found to trigger the disintegration and aggregation of polypeptides, thus changing protein solubility. The reasons for the increased solubility of proteins can be the deamination (removal of amide group) of polypeptides throughout the irradiation process and conversion from anti-water binding to water binding following the change in the amide group to the acidic group [45,46].

A similar pattern of results was observed in previous studies. Makari et al. (2021) [6] found an increase in the TSP of pistachio nuts after exposure to γ-rays (0, 0.5, 1, 1.5, 2, 4, and 6 kGy). Li et al. (2019) [44] found that the solubility of rice proteins was enhanced by increasing the e-beam dose. Malik et al. (2017) [45] also detected a rise in the solubility of sunflower seed proteins after γ-irradiation. Contrary to our results, Alinezhad et al. (2021) [22] found that the TSP of pistachio nuts decreased to the highest extent with the increase in the γ-radiation dose (≥1 kGy). Afify et al. (2011) [46] detected a decrease in the protein solubility of sesame, peanut, and soybean after γ-irradiation (0.5, 1, 2, 3, 5, and 7.5 kGy). In another study, a decline was observed in the protein solubility of two varieties of soybeans after γ-irradiation (0.25, 0.5, and 1 kGy). γ-rays reduced protein solubility by cross-linking protein chains and increasing hydrophobic interactions [23].

### 3.8. Proteins Profile of E-Beam Irradiated Pistachio Nuts

The protein profile of pistachios irradiated at e-beam doses of 0, 1, 2, 4, and 6 kGy are shown in Figure 6 and Figure 7. Figure 8 illustrates the densitometric analysis of control samples at 0.1 X, 0.5 X, and 0.25 X concentrations under Native-PAGE (a,b) and SDS-PAGE conditions (c,d). Figure 6a,b displays the protein profile of irradiated pistachios in native-PAGE and SDS-PAGE conditions, respectively. In Native-PAGE conditions, proteins maintain their original structure and do not undergo denaturation. With this method, the proteins were separated based on their size and electrical charge. The molecular weight of proteins in the Native-PAGE condition ranged from 7 to 310 kDa. In the SDS-PSGE condition, proteins were denatured in the presence of sodium dodecyl sulfate and separated based on their molecular weight. The molecular weight of proteins in the presence of sodium dodecyl sulfate was 0.862 to 297 kDa. Based on the results, the protein profile of the irradiated pistachios changed at a dose of 2 kGy (Figure 6a,b). The results of densitometry in both native-PAGE and SDS-PSGE conditions revealed that the intensity of protein bands with a molecular weight of 11 to 17 kDa increased at a dose of 2 kGy, while at this radiation dose, the intensity of protein bands with a molecular weight of 20 kDa and above was remarkably reduced (Figure 7a–d). Prior results of protein solubility (TSP) indicated that the solubility of proteins increased at a dose of 2 kGy; this elevation can be due to the stimulation of the activity of protease enzymes at this irradiation dose and the breaking of proteins with a high molecular weight due to increased amounts of protein units with low molecular weights. Other irradiation doses raised the protein band intensity compared to that of the control.

Makari et al. (2021) [6] reported an increase in the intensity of treated pistachio protein bands at a dose of 2 kGy, and a decrease in protein intensity when increasing the γ-radiation dose up to 6 kGy. Alinezhad et al. (2021) [22] reported that γ-rays changed the pattern of pistachio proteins such that the intensity of the bands decreased via γ-irradiation (6 kGy), while the protein bands with a molecular weight of 7 kDa at a dose of 4 kGy were intensified. Naei et al. (2019) [47] observed a reduction in the disappearance intensity of protein bands after the irradiation of pistachio nut extracts with high doses of γ-rays (100 kGy). According to Krishnan et al. (2015) [23], after γ-irradiation of two varieties of soybean, some protein bands disappeared or new bands appeared; they stated that these changes in band intensities can be explained by variations in the physicochemical properties of proteins such as oxidation, leading to the polymerization, condensation, or aggregation of proteins.

### 3.9. Malondialdehyde Content (MDA)

Table 2 lists the MDA values of un-irradiated and e-beam-irradiated pistachios (0, 1, 2, 4, and 6 kGy). The findings illustrate that the values of MDA in all samples remarkably increased with the increase in the e-beam irradiation dose (*p* < 0.05). The highest quantity of MDA was detected at doses higher than 4 kGy (Table 2).

These results are in line with those of previous studies. This finding supports previous research reporting a similar rising trend of MDA values in γ-irradiated pistachio nuts (0.5, 1, 1.5, 2, 4, and 6 kGy) [6]. In two separate studies, Mexis and Kontominas (2009) [20,30] reported an upward trend of the peroxide value in γ-irradiated (1, 1.5, 3, 5, and 7 kGy) pistachio, peanut, and cashew nut samples. The peroxide value of γ-irradiated hazelnut, walnut, almond, and pistachio nuts was reportedly raised as the γ-ray dose increased (1, 3, 5, and 7 kGy) [24]. Gölge and Ova (2008) [41] observed that γ-rays (0.5, 1, 3, and 5 kGy) enhanced the peroxide value of pine nut kernels. According to Sirisoontaralak and Noomhorm (2005) [48], a negligible increase was found in the TBA value of rice after γ-irradiation. Boonchoo et al. (2005) [10] also reported that γ-irradiation (6 kGy) raised the TBA value of brown rice. In contrast, Güler et al. (2017) [43] reported that the peroxide value of γ-irradiated hazelnuts remained unchanged after irradiation (0.5, 1, and 1.5 kGy). Lipid oxidation occurs due to the interaction of fatty acids with ionizing irradiation and the various reactions undergone causing oxidation, dehydration, polymerization, and decarboxylation [24]. Hydroperoxide decomposition into secondary products, including aldehydes, hydrocarbons, hydroxyl acids, ketones, and alcohols could adversely impact the sensory characteristics (color, odor, flavor, and crispiness) of irradiated nuts and reduce their overall acceptability [43]. Moreover, the oxidation of pistachio nuts’ predominant fatty acids, inclusive of oleic acid, linoleic acid, and palmitic acid, to peroxides and carbonyl compounds such as hexanal, pentanal, propanal, and acetaldehyde results in an unpleasant aroma and taste in irradiated pistachio nuts [48].

### 3.10. Profile of Fatty Acids

Table 3 presents the fatty acid composition of un-irradiated and irradiated pistachio kernel oil. The fatty acid composition of irradiated pistachio nuts was remarkably affected by higher doses of e-beam irradiation (*p* < 0.05). Oleic acid (61.34%), linoleic acid (22.10%), and palmitic acid (11.21%) are known as the most common monounsaturated fatty acids in pistachio nuts [20,22,24]. Table 3 showed an increase in the amount of saturated fatty acids (from 13.45 to 18.83) at e-beam doses ≥ 4 kGy (*p* < 0.05). Nevertheless, the differences in the percentage of saturated fatty acids were not significant at e-beam doses ≤ 2 kGy. The amounts of monounsaturated fatty acids (from 63.34 to 60.26) and polyunsaturated fatty acids (from 22.66 to 19.19) declined with an increasing radiation dose of up to 6 kGy (0.05 > *p*). The contents of monounsaturated and polyunsaturated fatty acids at doses ≤ 4 kGy remained unchanged. A remarkable decline in the amount of monounsaturated fatty acids including palmitoleic and oleic acids, and polyunsaturated fatty acids including linoleic and linolenic acids, was detected at high doses of e-beam (4 and 6 kGy) compared to the amounts of those in un-irradiated pistachios. The amount of other unsaturated fatty acids remained unchanged in e-beam-treated samples. Based on Table 3, 18-carbon fatty acids were affected by e-beam irradiation and diminished in content. The highest decrease was observed in oleic, linoleic, and linolenic acids, whereas the amounts of fatty acids, including stearic, palmitic, and margaric acids, increased after e-beam irradiation. However, irradiation had no impact on the contents of myristic, arachidic, and behenic acids, and these saturated fatty acids remained untouched after irradiation in all treatments (*p* < 0.05).

The results of the irradiated pistachio nuts’ fatty acid profile comply with those reported by Alinezhad et al. (2021) [22], who found that γ-rays at a dose ≥ 1 kGy increased the content of saturated fatty acids but reduced the amount of unsaturated fatty acids. γ-rays (1, 1.5, 3, 5, and 7 kGy) led to an increase in saturated fatty acids and a decrease in unsaturated fatty acids in pistachio nuts, peanuts, and cashew nuts [20,30]. Gecgel et al. (2011) [24] reported the same results, showing that γ-irradiation (1, 3, 5, and 7 kGy) raised the content of saturated fatty acids and lowered that of unsaturated fatty acids in pistachios, almonds, walnuts, and hazelnuts. In contrast, Gölge and Ova (2008) [41] did not detect any remarkable alteration in the fatty acid composition of γ-irradiated pine nut kernels and reported that palmitic, stearic, oleic, and linoleic acids remained unchanged after the treatment (0.5, 1, 3, and 5 kGy).

### 3.11. Sensory Evaluation

Figure 9 represents the scores of the sensory evaluation of un-irradiated and e-beam-irradiated pistachio nuts. According to the results, the panelists did not observe any alteration in the appearance of the treated pistachio nuts at doses ≤ 2 kGy. Still, the pistachios irradiated at doses ≥ 4 kGy received a lower score than the control did due to the darkening of the samples (*p* < 0.05). These results align with the color measurement, showing that the pistachio nuts’ color darkened after e-beam irradiation at doses up to 6 kGy. The taste of irradiated pistachios at doses ≤ 2 kGy received higher scores than the control did. On the other hand, samples irradiated at higher doses of e-beam (≥4 kGy) received lower scores than the control did. The texture of the samples affected by the e-beam became significantly more fragile than that of the control. Irradiation at a dose of 6 kGy significantly increased the texture fragility of the pistachios. E-beam treatment up to 4 kGy did not affect the overall acceptability of pistachio samples, while higher doses of irradiation (6 kGy) decreased the overall acceptability by lessening the quality features of the pistachio nut samples.

Similarly, Alinezhad et al. (2021) [22] reported that the quality attributes (color, taste, and texture) of γ-irradiated pistachio nuts were adversely affected by γ-irradiation at a dose of 6 kGy. Mexis and Kontominas (2009) [20] reported that γ-irradiation at doses > 3 kGy negatively impacted the sensory characteristics (color, texture, taste, and odor) of pistachio nuts and peanuts. Conversely, Sánchez-bel et al. (2008) [42] found that γ-irradiation up to 7 kGy did not negatively alter the sensory attributes of almonds. Gölge and Ova (2008) [41] observed no change in the sensory properties (texture and flavor) of walnuts immediately after γ-irradiation (0.5, 1, 1.5 kGy). It was also reported that the sensory attributes of hazelnuts, including color, flavor, crispiness, rancidity, and off-odor remained unaltered after γ-treatment (0.5, 1, and 1.5 kGy) [43].

## 4. Conclusions

The present research examined the effect of different doses of e-beam treatment (0, 1, 2, 4, and 6 kGy) on *A. flavus* contamination, reduction of AFB1 concentration, and the quality features of pistachios. The findings indicated that e-beam irradiation at doses ≥ 4 kGy effectively decreased the population of *A. flavus* spores by 4.620 log cycles and remarkably decreased the levels of AFB1 in pistachios. The maximum decreases were 99.69% and 100% at doses of 4 and 6 kGy, respectively. Three degradation products of AFB1, including C1, C2, and C3, were identified, and their structures and possible degradation pathways were presented. The TPC of pistachios increased at an e-beam dose of 2 kGy, and AOA declined in all samples in both DPPH and ABTS assays. Chlorophylls and carotenoids decreased in content due to higher doses of irradiation, and the color indices of pistachio nuts changed, leading to a darker color. TSP increased, and a change in the protein profile of the treated pistachios was detected at a dose of 2 kGy. Moreover, e-beam raised the amount of MDA and altered the profile of fatty acids at a dose of 6 kGy, increasing the amount of saturated fatty acids and reducing the amount of unsaturated fatty acids in the pistachio nuts. The sensory characteristics of pistachio kernels were negatively affected, thereby lowering the overall acceptability of pistachio nuts at a dose of 6 kGy. The results suggested that e-beam at doses ≥ 4 kGy can effectively remove *A. flavus* contamination from pistachio nuts and reduce the AFB1 concentration. However, e-beam dose ≤ 2 kGy maintained the quality attributes of pistachio nuts.

## Figures and Tables

**Figure 1 foods-12-03040-f001:**
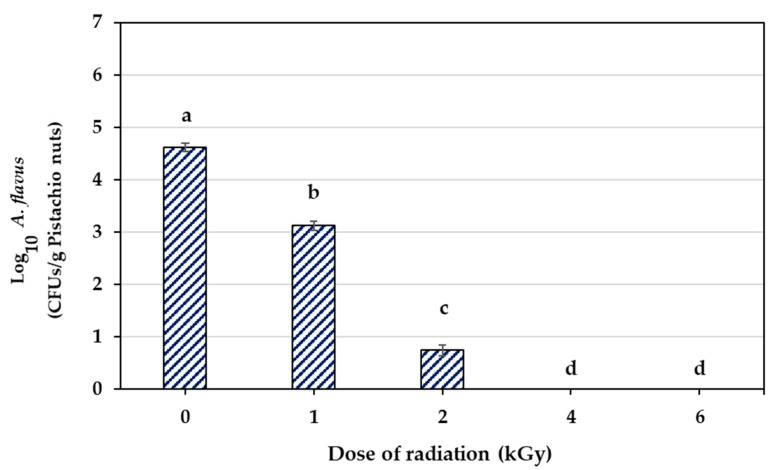
Disinfection of *A. flavus* using different doses of e-beam radiation on infected pistachios. Data display the means of triplicates. Different letters above bars show significant differences between treatments (*p* < 0.05, based on Duncan’s test).

**Figure 2 foods-12-03040-f002:**
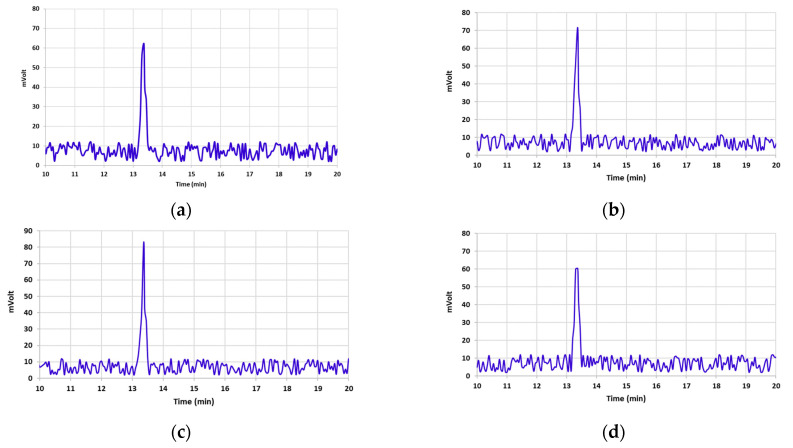
Chromatograms obtained via analysis of HPLC illustrate AFB1 detoxification of contaminated pistachio kernels at different doses of e-beam irradiation (0, 1, 2, 4, and 6 kGy (**a**–**e**)).

**Figure 3 foods-12-03040-f003:**
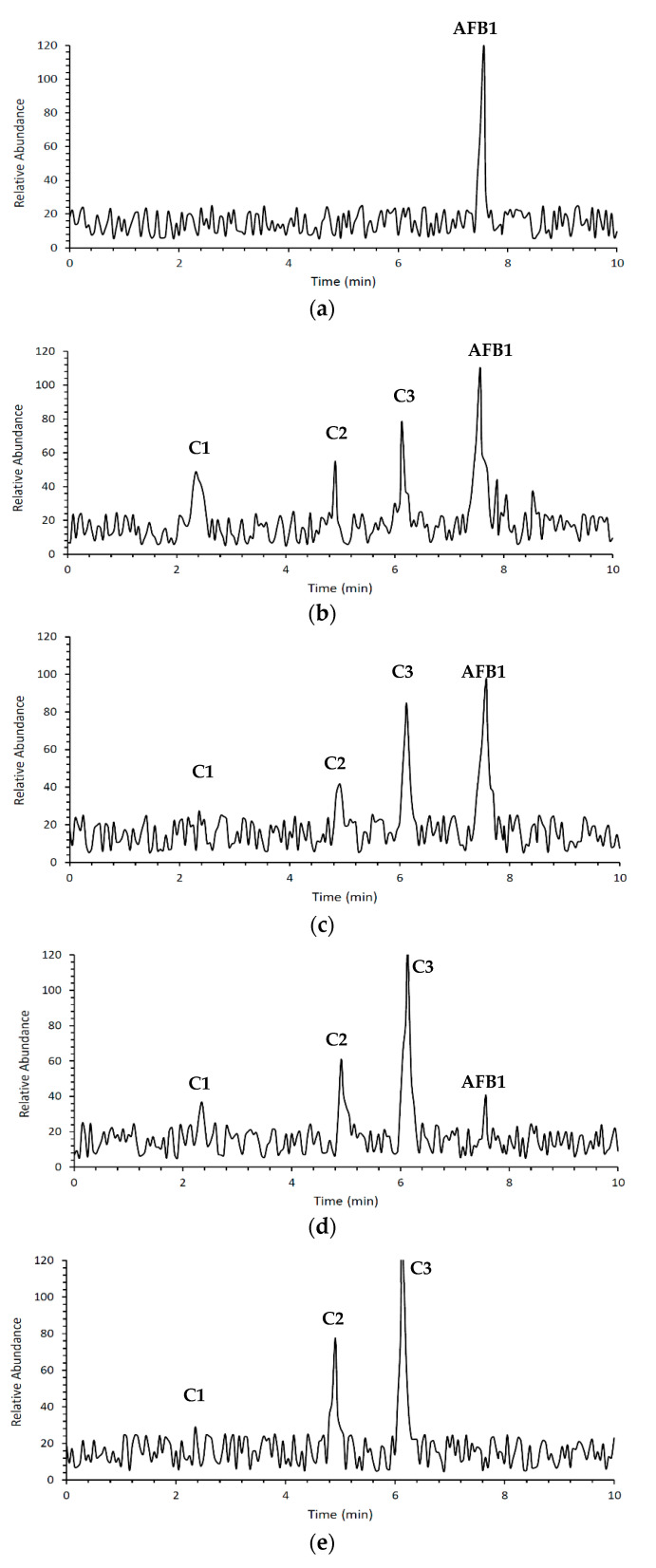
Chromatograms of contaminated pistachio kernels with AFB1 at different doses of e-beam irradiation (0, 1, 2, 4, and 6 kGy (**a**–**e**)) according to LC analysis.

**Figure 4 foods-12-03040-f004:**
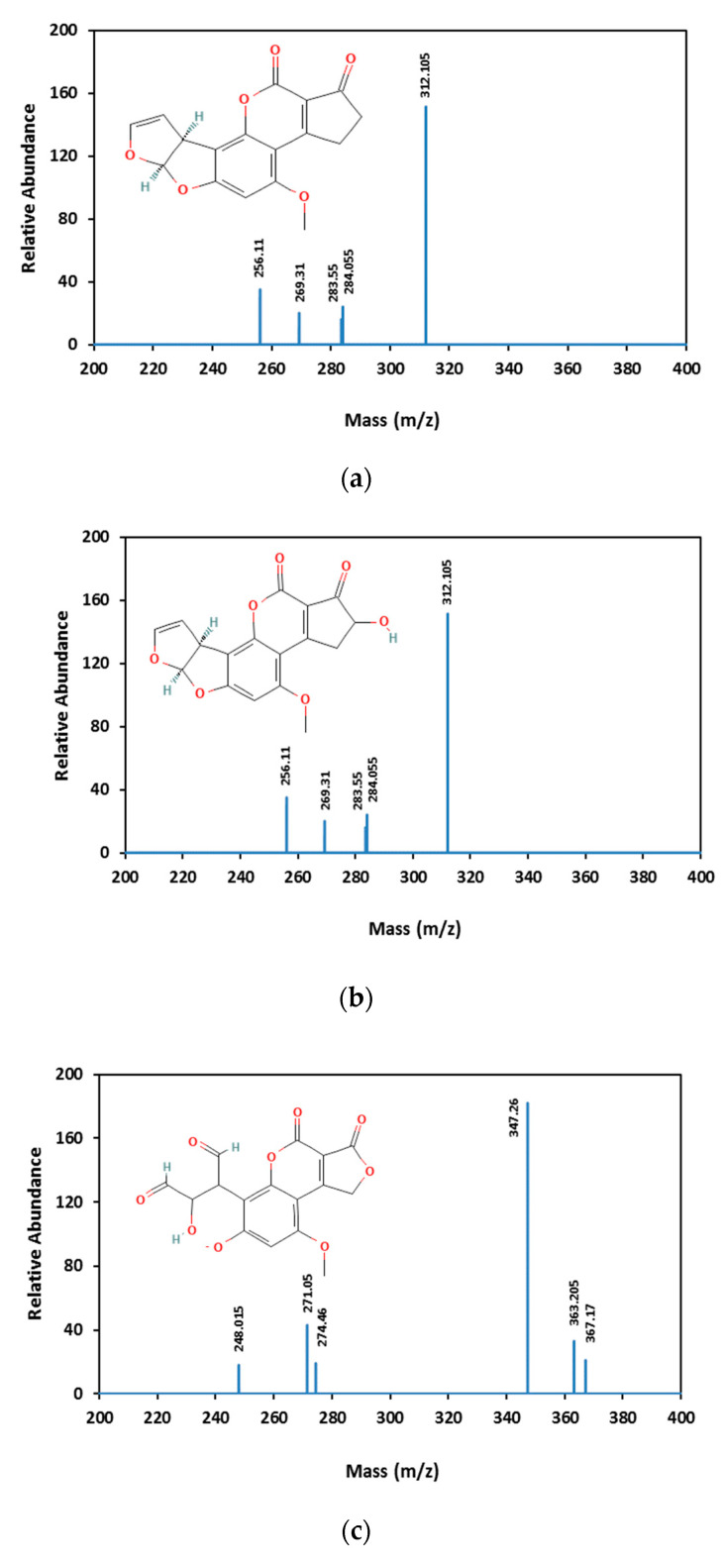
Mass spectrometric analysis of AFB1 and its degradation products. AFB1 (**a**) and the degradation products of AFB_1_ (**b**–**d** are C1, C2, and C3, respectively) analyzed using MS.

**Figure 5 foods-12-03040-f005:**
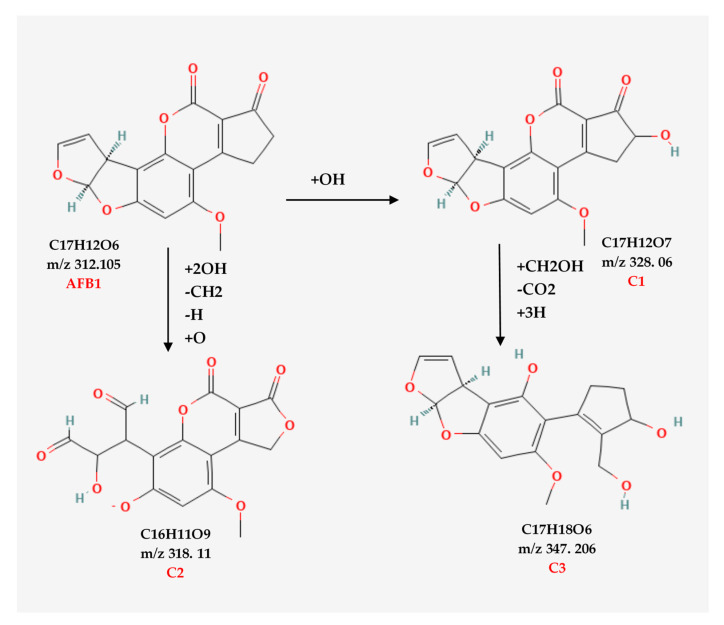
The possible structures related to the degraded products and possible degradation pathways of AFB1.

**Figure 6 foods-12-03040-f006:**
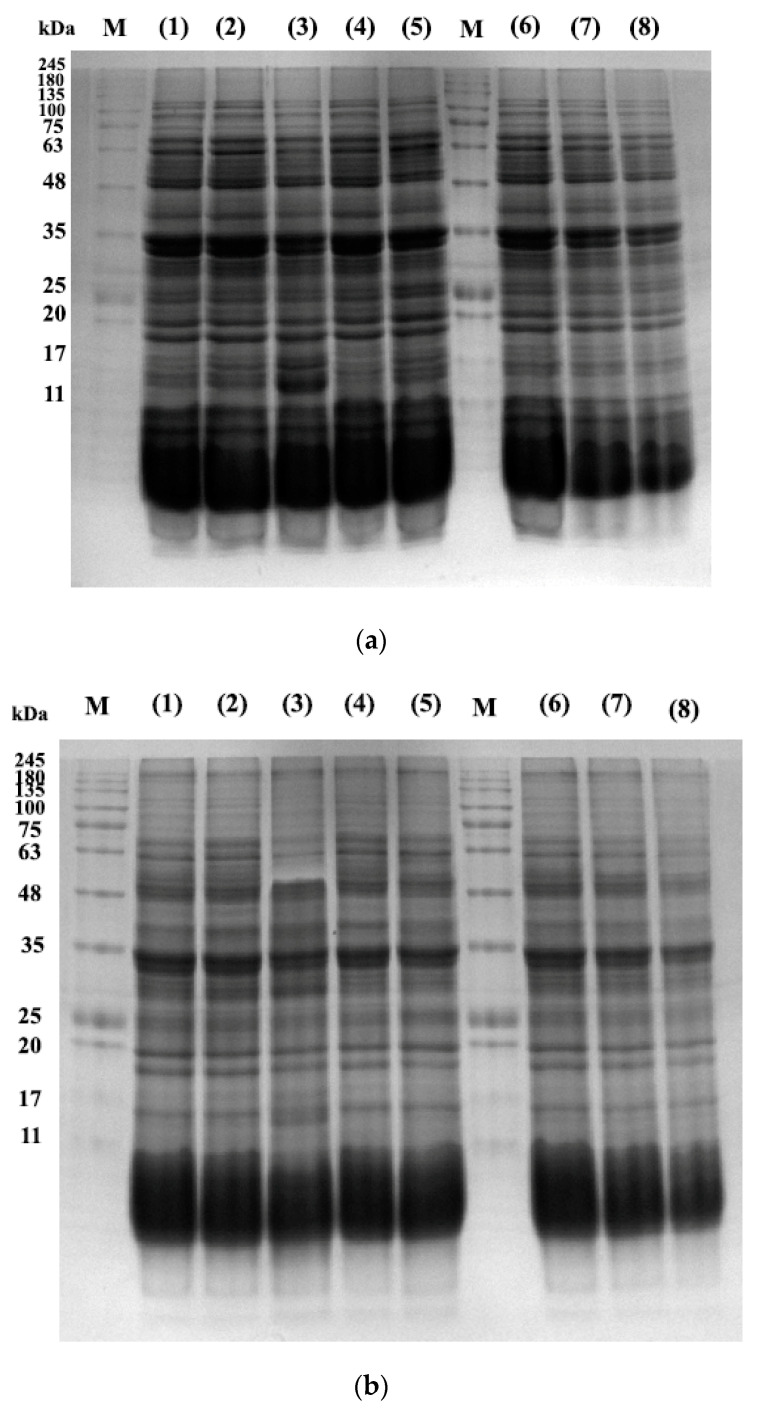
Native-PAGE (**a**) and SDS-PAGE (**b**) profiles of the control sample (lane 1) and e-beam treated pistachios proteins (lane numbers 2, 3, 4, and 6 are 1, 2, 4, and 6 kGy, respectively). Lane numbers 6, 7, and 8 show controls in 0.1 X, 0.5 X, and 0.25 X concentrations. “M” indicates a molecular weight marker (SinaClon Bio-Science, Prestained protein ladder, PR901641).

**Figure 7 foods-12-03040-f007:**
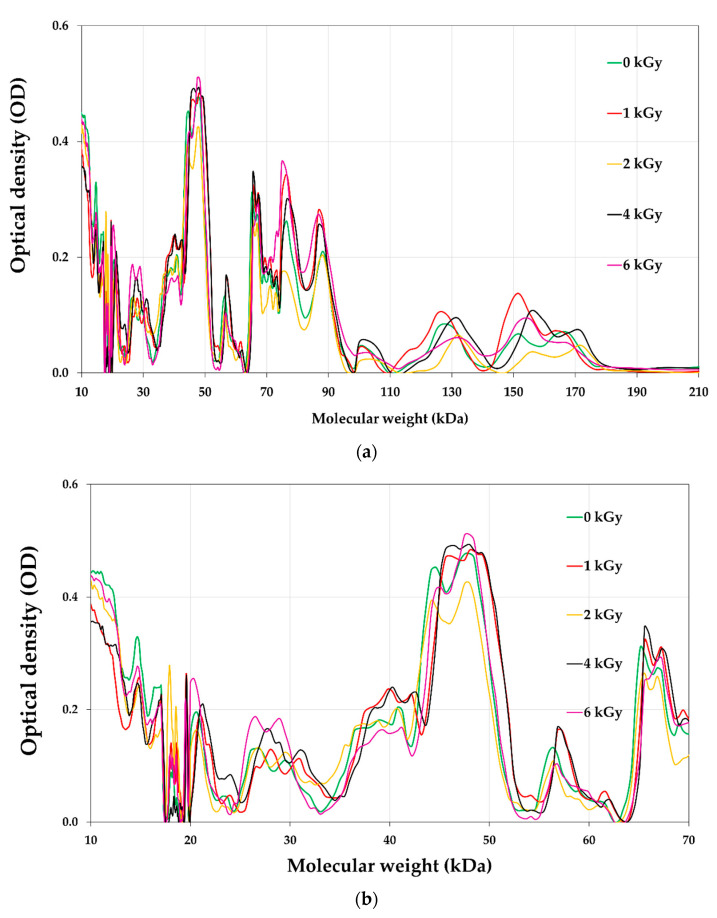
Densitometry analysis of Native-PAGE (**a**,**b**) and SDS-PAGE (**c**,**d**) profiles of soluble proteins extracted from un-irradiated and e-beam-irradiated (1, 2,4, and 6 kGy) pistachio nut kernels.

**Figure 8 foods-12-03040-f008:**
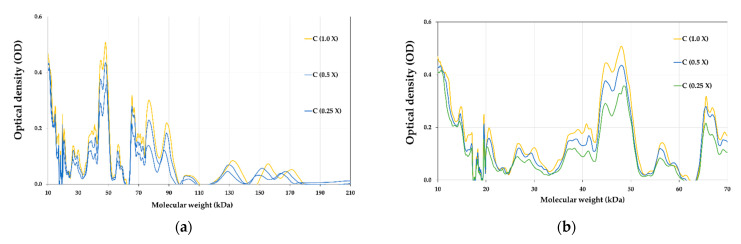
Densitometric analysis of control samples in 0.1 X, 0.5 X, and 0.25 X concentrations under Native-PAGE (**a**,**b**) and SDS-PAGE (**c**,**d**) conditions.

**Figure 9 foods-12-03040-f009:**
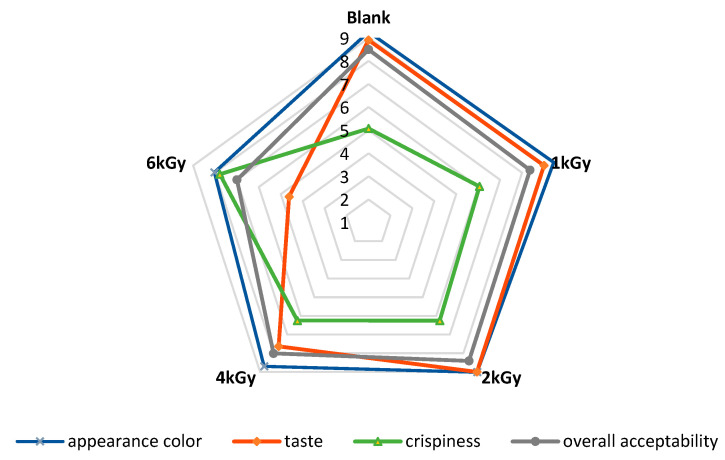
Sensory properties of un-irradiated and irradiated pistachio nuts as a function of irradiation dose.

**Table 1 foods-12-03040-t001:** Effect of different doses of e-beam irradiation (0, 1, 2, 4, and 6 kGy) on AFB1 (ppb) levels in infected pistachio kernels.

	Dose of Irradiation (k Gy)				
	0	1	2	4	6
Aflatoxin concentration (ppb)	692.1 ± 0.12 ^a^	546.2 ± 0.14 ^b^	168.5 ± 0.21 ^c^	2.10 ± 0.08 ^d^	0.0 ± 0.0 ^d^

Mean ± standard deviation with different letters in the same row differ statistically at the 0.05 probability level according to Duncan’s test.

**Table 2 foods-12-03040-t002:** Total phenolic content (TPC), antioxidant activity (AOA), total soluble protein (TSP), malondialdehyde (MDA), pigments, and instrumental color indices of pistachios after e-beam irradiation (at 1, 2, 4, and 6 kGy).

Properties	Dose of E-Beam Irradiation (kGy)				
	0	1	2	4	6
TPC (mg GAE/g)	6.61 ± 0.10 ^bc^	6.48 ± 0.17 ^c^	7.65 ± 0.36 ^a^	6.85 ± 0.38 ^bc^	7.02 ± 0.15 ^b^
Inhibitory of DPPH activity (1 g/30 mL, %)	82.07 ± 1.45 ^d^	81.06 ± 0.95 ^b^	78.17 ± 1.28 ^a^	78.74 ± 1.05 ^c^	77.21 ± 0.73 ^e^
Inhibitory of ABTS activity (1 g/30 mL, %)	78.85 ± 0.42 ^a^	78.92 ± 0.21 ^a^	77.28 ± 0.26 ^b^	77.72 ^b^ ± 0.13 ^b^	76.03 ± 0.13 ^c^
TSP (mg/g)	1.59 ± 0.11 ^c^	1.81 ± 0.07 ^b^	2.10 ± 0.04 ^a^	1.67 ± 0.03 ^c^	1.71 ± 0.04 ^bc^
MDA (n mol. g^−1^)	7.01 ± 0.15 ^e^	7.61 ± 0.20 ^d^	8.78 ± 0.29 ^c^	11.00 ± 0.23 ^b^	12.18 ± 0.34 ^a^
Pigments					
Chlorophyll a	17.15 ± 0.93 ^a^	12.99 ± 1.00 ^b^	13.47 ± 0.87 ^b^	10.55 ± 0.53 ^c^	9.07 ± 0.28 ^d^
Chlorophyll b	22.40 ± 2.30 ^a^	22.73 ± 0.71 ^a^	22.41 ± 0.72 ^a^	12.88 ± 1.10 ^b^	14.24 ± 0.77 ^b^
Total chlorophyll	39.54 ± 3.21 ^a^	35.72 ± 1.28 ^b^	35.88 ± 0.32 ^b^	23.43 ± 1.65 ^c^	23.31 ± 0.91 ^c^
Total carotenoid	19.33 ± 0.65 ^a^	13.04 ± 0.54 ^d^	15.12 ± 0.90 ^b^	14.39 ± 0.60 ^bc^	13.96 ± 0.36 ^cd^
Instrumental color					
Lightness (L)	59.03 ± 1.19 ^a^	58.76 ± 0.72 ^a^	58.75 ± 0.61 ^a^	57.39 ± 0.59 ^b^	55.18 ± 1.09 ^c^
Redness (a*)	−5.11 ± 0.16 ^b^	−5.07 ± 0.15 ^b^	−5.07 ± 0.11 ^b^	−4.93 ± 0.05 ^b^	−4.12 ± 0.09 ^a^
Yellowness (b*)	29.36 ± 0.89 ^a^	29.55 ± 0.48 ^a^	29.53 ± 0.41 ^a^	27.35 ± 0.63 ^b^	27.43 ± 0.96 ^b^
ΔE	00.00 ± 00.00 ^d^	0.33 ± 0.62 ^c^	0.33 ± 0.75 ^c^	2.60 ± 0.66 ^b^	4.42 ± 0.14 ^a^

Mean ± standard deviation with different letters in the same row differ statistically (*p* < 0.05).

**Table 3 foods-12-03040-t003:** Fatty acid composition (%) of un-irradiated and e-beam-irradiated pistachios.

	X-ray Irradiation Dose (kGy)				
Compounds	0 (Blank)	1	2	4	6
Myristic acid (14:0)	0.16 ± 0.03 ^a^	0.14 ± 0.02 ^a^	0.14 ± 0.02 ^a^	0.13 ± 0.01 ^a^	0.15 ± 0.01 ^a^
Myristoleic acid (14:1)	0.01 ± 0.01 ^a^	0.03 ± 0.00 ^a^	nd ^b^	nd ^b^	nd ^b^
Pentadecanoic acid (15:0)	0.02 ± 0.01 ^a^	0.04 ± 0.01 ^a^	0.03 ± 0.02 ^a^	0.04 ± 0.02 ^a^	0.05 ± 0.02 ^a^
Palmitic acid (16:0)	11.21 ± 0.15 ^c^	11.32 ± 0.30 ^c^	11.38 ± 0.06 ^c^	12.17 ± 0.32 ^b^	14.58 ± 0.48 ^a^
Palmitoleic acid (16:1)	1.16 ± 0.09 ^ab^	1.15 ± 0.07 ^ab^	1.16 ± 0.12 ^ab^	1.09 ± 0.14 ^b^	1.27 ± 0.05 ^a^
Margaric acid (17:0)	0.02 ± 0.01 ^b^	0.03 ± 0.00 ^ab^	0.02 ± 0.01 ^b^	0.05 ± 0.00 ^a^	0.02 ± 0.01 ^ab^
Stearic acid (18:0)	1.78 ± 0.31 ^b^	1.59 ± 0.08 ^b^	1.86 ± 0.17 ^b^	2.97 ± 0.31 ^a^	3.68 ± 0.84 ^a^
Oleic acid (18:1)	61.34 ± 0.17 ^a^	61.73 ± 0.34 ^a^	61.77 ± 0.19 ^a^	60.53 ± 0.58 ^a^	58.16 ± 0.99 ^b^
Linoleic acid (18:2)	22.10 ± 0.37 ^a^	21.92 ± 0.12 ^a^	21.86 ± 0.25 ^a^	21.47 ± 0.16 ^a^	18.75 ± 0.62 ^b^
Linolenic acid (18:3)	0.56 ± 0.03 ^a^	0.58 ± 0.02 ^a^	0.57 ± 0.03 ^a^	0.55 ± 0.05 ^a^	0.44 ± 0.07 ^b^
Arachidic acid (20:0)	0.28 ± 0.01 ^a^	0.26 ± 0.01 ^a^	0.23 ± 0.02 ^a^	0.26 ± 0.07 ^a^	0.25 ± 0.04 ^a^
Gondoic acid (20:1)	0.83 ± 0.01 ^a^	0.79 ± 0.07 ^a^	0.83 ± 0.08 ^a^	0.82 ± 0.07 ^a^	0.82 ± 0.05 ^a^
Behenic acid (22:0)	0.07 ± 0.03 ^a^	0.08 ± 0.02 ^a^	0.07 ± 0.02 ^a^	0.06 ± 0.01 ^a^	0.08 ± 0.02 ^a^
ƩMUFA	63.34 ± 0.06 ^a^	63.68 ± 0.34 ^a^	63.76 ± 0.23 ^a^	62.45 ± 0.53 ^ab^	60.26 ± 1.05 ^c^
ƩPUFA	22.66 ± 0.31 ^a^	22.51 ± 0.11 ^a^	22.44 ± 0.25 ^a^	22.00 ± 0.11 ^a^	19.19 ± 0.62 ^b^
ƩUSFA	86.00 ± 0.38 ^a^	86.19 ± 0.37 ^a^	86.21 ± 0.10 ^a^	84.53 ± 0.63 ^b^	79.45 ± 1.64 ^c^
ƩSFA	13.45 ± 0.42 ^c^	13.41 ± 0.27 ^c^	13.74 ± 0.24 ^c^	15.68 ± 0.38 ^b^	18.83 ± 1.08 ^a^

nd, non-detected. Mean ± standard deviation values with different superscript letters (a, b, and c) in the same row differ statistically (*p* < 0.05).

## Data Availability

The datasets generated for this study are available on request to the corresponding author.

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
