# Peer review of "Use of X-Irradiations in Reducing the Waste of Aflatoxin-Contaminated Pistachios and Evaluation of the Physicochemical Properties of the Irradiated Product"

_foods, 2023, doi:10.3390/foods12163040_

Round 1

Reviewer 1 Report

Using of the X-irradiation in reducing the waste of aflatoxin contaminated pistachios and evaluation of physicochemical properties of irradiated product, this article has been well written, but some things need to be completed:   1. The introduction regarding the urgency and novelty of this study needs to be sharpened. 2. Figure 2 should be explained more fully in terms of the chromatogram. 3. In Table 2, it is necessary to add the total color change (ΔE) before and after irradiation because this can provide very important information. Also, in the method and discussion related to ΔE 4. In Table 2, related to the Inhibitory of DPPH activity and ABTS, please inform the inhibition at what sample concentration? 5. Overall, in the discussion section, it is necessary to sharpen the discussion in more detail and comprehensively on each phenomenon of the data obtained and the figures shown.

Moderate editing of English language required

Author Response

Response to Respected Reviewer #1 Comments

Using of the X-irradiation in reducing the waste of aflatoxin contaminated pistachios and evaluation of physicochemical properties of irradiated product, this article has been well written, but some things need to be completed: 

Thanks for your positive comments. The corrections you want are marked with red color in the revised manuscript.

Point 1 The introduction regarding the urgency and novelty of this study needs to be sharpened.

Response 1: Done as suggested.

Point 2: Figure 2 should be explained more fully in terms of the chromatogram. 

Response 2: Done as suggested.

Point 3: In Table 2, it is necessary to add the total color change (ΔE) before and after irradiation because this can provide very important information. Also, in the method and discussion related to ΔE 

Response 3: Done as suggested.

Point 4: In Table 2, related to the Inhibitory of DPPH activity and ABTS, please inform the inhibition at what sample concentration? 

Response 4: Done as suggested.

Point 5: Overall, in the discussion section, it is necessary to sharpen the discussion in more detail and comprehensively on each phenomenon of the data obtained and the figures shown.

Response 5: Done as suggested.

Point 6: Comments on the Quality of English Language: Moderate editing of English language required

Response 6: The editing of English language of the manuscript was done by a reputable English Academy (Virayeh), whose certificate has also been added below.

Reviewer 2 Report

Dear authors,

The topic is  very interesting and within the scope of this Journal, I want you to congratulate for it.

 Keywords: I think it would be more advisable to include the terms of pistachio and aflatoxin.

Introduction:

 It would be interesting to discuss in the introduction the authorisation of this technique from a legal point of view.

I am not an expert in the technique and I would like to know the differences between γ-rays, X-rays and e- beam.. and the effect of each on what is to be evaluated. In the introduction a lot of importance is given to the sanitary part but not to the other organoleptic and functional part. I think that everything that is going to be discussed should have bibliographical support in the introduction.

I would like to know if this is feasible or has already been done in a nut company.

I would also like to detail the effect on moulds (A. flavus) and aflatoxins (AFB1 and B2) so that the reader can see the effectiveness of this technique in previous publications.

Reference 5: The newest information from “EC Regulation No 2023/915 of 25 April 2023 setting maximum levels for certain contaminants in foodstuffs and repealing Regulation (EC) No 1881/2006 of 19 December 2006” is necessary to be now taken into account.

Materials and methods

It should detail more precisely how sampling is done.

I would like to know why you have decided to do the experiment only on AFB1 and not on B2 as A flavus can produce both.

Results and discussion

Line 292: aflatoxin should be written in lower case.

Best regards

Author Response

Response to Respected Reviewer #2 Comments

Dear authors,

The topic is very interesting and within the scope of this Journal, I want you to congratulate for it.

Thanks for your positive comments. The corrections you want are marked with purple color in the revised manuscript.

Point 1: Keywords: I think it would be more advisable to include the terms of pistachio and aflatoxin.

Response 1: Done as suggested.

Point 2: It would be interesting to discuss in the introduction the authorisation of this technique from a legal point of view.

Response 2: Done as suggested.

Point 3: I am not an expert in the technique and I would like to know the differences between γ-rays, X-rays and e- beam, and the effect of each on what is to be evaluated. In the introduction a lot of importance is given to the sanitary part but not to the other organoleptic and functional part. I think that everything that is going to be discussed should have bibliographical support in the introduction.

Response 3: Done as suggested.

Point 4: I would like to know if this is feasible or has already been done in a nut company.

Response 4: As explained in the introduction, the irradiation technique has been used for many years to treat foodstuffs and nuts by different countries, and access to irradiation facilities is possible for nut companies. But until today, the use of this technique has not been reported by a particular nut company.

Food irradiation has been extensively used in Asia and the United States [14]; however, the EU has reduced commercial food irradiation after the EU directive 1999/2/EC on the approximation of the laws of the member states concerning foods and food ingredients treated with ionizing radiation [15]. Moreover, EU regulation 1999/3/EC permitted the application of ionizing irradiation to a few food commodities comprising dried aromatic herbs, spices, and vegetable seasonings [16]. Nonetheless, EU members are still allowed to utilize irradiation for different types of foodstuffs, especially nuts [17].

Point 5: I would also like to detail the effect on moulds (A. flavus) and aflatoxins (AFB1 and B2) so that the reader can see the effectiveness of this technique in previous publications.

Response 5: Done as suggested.

Point 6: Reference 5: The newest information from “EC Regulation No 2023/915 of 25 April 2023 setting maximum levels for certain contaminants in foodstuffs and repealing Regulation (EC) No 1881/2006 of 19 December 2006” is necessary to be now taken into account.

Response 6: This reference was updated.

Point 7: It should detail more precisely how sampling is done.

Response 7: Done as suggested.

Point 8: I would like to know why you have decided to do the experiment only on AFB1 and not on B2 as A flavus can produce both.

Response 8: The reason for choosing AFB1 in this study is mentioned in the introduction.

The most common aflatoxins produced by toxigenic strains of A. flavus are aflatoxin B1 (AFB1) and aflatoxin B2 (AFB2) [2]. AFB1 is more dangerous than AFB2 and other aflatoxins. Due to its potent carcinogenic, hepatotoxic, teratogenic, and immunosuppressive impacts on humans, it is classified as a Group 1 carcinogen by the International Agency for Research on Cancer (IARC) [1,4]. The major genotoxic and carcinogenic metabolite of AFB1 is believed to be aflatoxin B1- exo-8,9-epoxide (AFBO), which is also metabolized into several oxidized metabolites such as hydroxylated products, including aflatoxin M1 (AFM1), aflatoxin Q1 (AFQ1), aflatoxin P1 (AFP1), and aflatoxicol (AFL). AFM1 is known to be the most carcinogenic hydroxylated metabolite of AFB1 [4].

Point 9: Line 292: aflatoxin should be written in lower case.

Response 9: Edited as suggested.

Reviewer 3 Report

General comment

This study presents the effects of X-rays on the viability of Aspergillus flavus and the aflatoxin content when pistachios are processed. The effects of X-rays on food have been known for many years, and the scientific contribution of the studies carried out on this subject is moderate. A number of points need to be reconsidered:

1- regulations concerning the decontamination of food intended for human consumption are imprecise or inaccurate

2- processes aimed at degrading toxins, particularly aflatoxins, must be accompanied by an assessment of the structure and toxicity of the degradation products formed

3- The importance of chemical analyses of foodstuffs must be clarified in the introduction: analyses of total composition are necessary but not sufficient. In this respect, explanations should be provided concerning non-specific oxidation measurements of foods and specific measurements of proteins (balance between reduced and oxidised forms, thiols and disulphide bonds), vitamins (water-soluble and fat-soluble), poly-unsaturated fatty acids, etc.

4- A number of clarifications are required in different parts of the manuscript: introduction, materials and methods, results and discussion: see detailed comments below.

Detailled comments

L22 : this last sentence is confusing regarding what was said above

L34: No, the genotoxicity of aflatoxins and their carcinogenic power is the major point of their toxicity, much more than their oxidising capacity.

L40: No, the chemical decontamination of food intended for human consumption is not permitted in Europe.

The structure and toxicity of metabolites generated during food processing must be assessed, particularly in the case of aflatoxins.

L120: add validation data

L122-170: these methods are global physico-chemical methods which do not prejudge the effects of treatments on essential compounds: sulphur amino acids, vitamins, polyunsaturated fatty acids, etc.

L171-179: Given the way MDA is produced, what is expected from this measure? This point should be explained in the introduction.

L180-200: The value of comparing fatty acid profiles before and after treatment should be explained in the introduction.

L218-260: OK, but as pointed out in the discussion, these results confirm numerous previous data. The presentation of the manuscript should highlight new points

L261-297: Radiation-induced degradation of AFB1 has been known for a very long time. What is needed now is a characterisation of the metabolites formed and an analysis of their toxicity.

L298-405: These results confirm numerous previous studies, but do not allow us to assess the changes observed at the molecular level. For example, what about the decrease in carotenoid content observed at the lowest dose: what are the mechanisms and consequences?

How can we think that a reduction in chlorophyll synthesis (L350) is possible under these experimental conditions? What about the toxicity of the degradation compounds formed (radical products)?

What links can be made between the above-mentioned changes and the change in colour?

L406-449: This part of the manuscript is too detailed for its intended purpose.

L450-476: Yes, and the introduction should focus on tests that reveal changes in the chemical composition of foods: general oxidation as measured by MDA levels, but also oxidation of proteins (R-SH and R-SS-R' balance), vitamins, fatty acids, etc.

L477-516: These analyses are perfectly justified, but the introduction fails to show why they are necessary. Similarly, the increase in MDA observed previously should give rise to questions/explanations as to its origin, particularly in the absence of any change in the rate of oxidation of polyunsaturated fatty acids.

Author Response

Response to Respected Reviewer #3 Comments

General comment

This study presents the effects of X-rays on the viability of Aspergillus flavus and the aflatoxin content when pistachios are processed. The effects of X-rays on food have been known for many years, and the scientific contribution of the studies carried out on this subject is moderate. A number of points need to be reconsidered:

Point 1: regulations concerning the decontamination of food intended for human consumption are imprecise or inaccurate.

Response 1: This part was corrected.

Point 2: processes aimed at degrading toxins, particularly aflatoxins, must be accompanied by an assessment of the structure and toxicity of the degradation products formed

Response 2: The assessment of AFB1 degradation products was done using LC/MS as suggested and was added to the manuscript in blue color.

Point 3: The importance of chemical analyses of foodstuffs must be clarified in the introduction: analyses of total composition are necessary but not sufficient. In this respect, explanations should be provided concerning non-specific oxidation measurements of foods and specific measurements of proteins (balance between reduced and oxidised forms, thiols and disulphide bonds), vitamins (water-soluble and fat-soluble), poly-unsaturated fatty acids, etc.

Response 3: Done as suggested.

Point 4: L22 : this last sentence is confusing regarding what was said above

Response 4: This sentence was cleared.

Point 5: L34: No, the genotoxicity of aflatoxins and their carcinogenic power is the major point of their toxicity, much more than their oxidising capacity.

Response 5: This part was corrected.

Point 6: L40: No, the chemical decontamination of food intended for human consumption is not permitted in Europe.

Response 6: This part was corrected.

Point 7: The structure and toxicity of metabolites generated during food processing must be assessed, particularly in the case of aflatoxins.

Response 7: Done as suggested.

Point 8: L120: add validation data

Response 8: Done as suggested.

Point 9: L122-170: these methods are global physico-chemical methods which do not prejudge the effects of treatments on essential compounds: sulphur amino acids, vitamins, polyunsaturated fatty acids, etc.

Response 9: Thanks for your comment. The ionizing radiation directly or indirectly affects food composition. Water radiolysis and production of free radicals such as oxygen, hydrogen, hydroxyl, and hydrogen peroxide radicals cause oxidation and reduction reactions in food and affect food constituents such as vitamins, proteins, unsaturated fatty acids, phenolic compounds, etc, leading to the reduction of food nutritional value [1]. Our purpose in conducting these tests was to investigate the general effect of electron beam irradiation on the physicochemical properties of pistachios. For this reason, general tests were considered to check these changes. However, the profile of fatty acids (unsaturated and saturated fatty acids) and the profile of proteins were investigated.

[1]Mostafavi, H. A., Mirmajlessi, S. M., & Fathollahi, H. (2012). The potential of food irradiation: Benefits and limitations. Trends in vital food and control engineering5, 43-68.

Point 10: L171-179: Given the way MDA is produced, what is expected from this measure? This point should be explained in the introduction.

Response 10: Done as suggested.

Point 11: L180-200: The value of comparing fatty acid profiles before and after treatment should be explained in the introduction.

Response 11: Done as suggested.

Point 12: L218-260: OK, but as pointed out in the discussion, these results confirm numerous previous data. The presentation of the manuscript should highlight new points

Response 12: Thanks for your comment.

Point 13: L261-297: Radiation-induced degradation of AFB1 has been known for a very long time. What is needed now is a characterisation of the metabolites formed and an analysis of their toxicity.

Response 13: Identification of aflatoxin degradation products after electron beam treatment was done and added to the manuscript in blue color.

Point 14: L298-405: These results confirm numerous previous studies, but do not allow us to assess the changes observed at the molecular level. For example, what about the decrease in carotenoid content observed at the lowest dose: what are the mechanisms and consequences?

How can we think that a reduction in chlorophyll synthesis (L350) is possible under these experimental conditions? What about the toxicity of the degradation compounds formed (radical products)?

What links can be made between the above-mentioned changes and the change in colour?

Response 14: These questions were answered and added to the results and discussion.

Point 15: L406-449: This part of the manuscript is too detailed for its intended purpose.

Response 15:  Thank you for your comment. It was done.

Point 16: L450-476: Yes, and the introduction should focus on tests that reveal changes in the chemical composition of foods: general oxidation as measured by MDA levels, but also oxidation of proteins (R-SH and R-SS-R' balance), vitamins, fatty acids, etc.

Response 16: Done as suggested.

Point 17: L477-516: These analyses are perfectly justified, but the introduction fails to show why they are necessary. Similarly, the increase in MDA observed previously should give rise to questions/explanations as to its origin, particularly in the absence of any change in the rate of oxidation of polyunsaturated fatty acids.

Response 17: Done as suggested.

Round 2

Reviewer 2 Report

Dear Authors,

thanks for your comments and clarifications.

There are a few things to highlight in the text:

Line 116: "Fungal" should be written in capital letters.

Line 291: Log 10 in the graphic is not ok

Reviewer 3 Report

Thank you for clarifying this revised version of your manuscript.